# Imbibition-induced selective wetting of liquid metal

Ji-Hye Kim[1,7], Sooyoung Kim[2,7], Hyeonjin Kim [3], Sanghyuk Wooh[3], Jiung Cho[4], Michael D. Dickey [5], Ju-Hee So [6] ✉ & Hyung-Jun Koo [2] ✉

Herein, we present the imbibition-induced, spontaneous, and selective wetting characteristics of gallium-based liquid metal alloys on a metallized surface with micro-scale topographical features. Gallium-based liquid metal alloys are fascinating materials that have enormous surface tension; therefore, they are difficult to pattern into films. The complete wetting of eutectic alloy of gallium and indium is realized on microstructured copper surfaces in the presence of HCl vapor, which removes the native oxide from the liquid metal alloy. This wetting is numerically explained based on the Wenzel's model and imbibition process, revealing that the dimensions of the microstructures are critical for effective imbibition-driven wetting of the liquid metal. Further, we demonstrate that the spontaneous wetting of the liquid metal can be directed selectively along the microstructured region on the metallic surface to create patterns. This simple process enables the uniform coating and patterning of the liquid metal over large areas without an external force or complex processing. We demonstrate that the liquid metal-patterned substrates maintain electrical connection even in a stretched state and after repetitive stretching cycles.

Gallium-based liquid metal alloys (GaLMs) are of significant interest owing to their attractive properties, such as low melting point, high electrical conductivity, low viscosity and fluidity, low toxicity, and high deformability[1,2]. Pure gallium has a melting point of ~30 °C, which falls below room temperature upon forming alloys with certain metals such as In and Sn at eutectic compositions. Two important GaLMs are eutectic gallium–indium alloy (EGaIn, 75% Ga and 25% In by weight, melting point: 15.5 °C) and eutectic gallium–indium–tin alloy (GaInSn or Galinstan, 68.5% Ga, 21.5% In, and 10% Sn by weight, melting point: -11 °C)[1,2]. Because they are electrically conductive in the liquid phase, GaLMs have been actively studied as a stretchable or deformable electronic pathway for various applications, including electronics[3–9],

strain or bending sensors[10–14], and conductive wires[15–17]. Fabricating such devices through the deposition, printing, and patterning of GaLMs requires understanding and controlling the interfacial characteristics of GaLMs with underlying substrates. GaLMs have high surface tension (624 mN m$^{-1}$ for EGaIn[18,19] and 534 mN m$^{-1}$ for Galinstan[20,21]), which could make them difficult to handle or manipulate. The formation of a solid native gallium oxide skin on the surface of GaLMs under ambient conditions provides a shell that stabilizes GaLMs in non-spherical shapes. This property has allowed GaLMs to be printed, injected into microchannels, and patterned by taking advantage of the interfacial stability enabled by the oxide[19,22–27]. The solid oxide shell also allows GaLMs to adhere to most smooth surfaces, but

[1]Department of Energy and Chemical Engineering, Seoul National University of Science & Technology, 232 Gongneung-ro, Nowon-gu, Seoul 01811, Republic of Korea. [2]Department of Chemical & Biomolecular Engineering, Seoul National University of Science & Technology, 232 Gongneung-ro, Nowon-gu, Seoul 01811, Republic of Korea. [3]School of Chemical Engineering & Materials Science, Chung-Ang University, 84 Heukseok-ro, Dongjak-gu, Seoul 06974, Republic of Korea. [4]Western Seoul Center, Korea Basic Science Institute, 150 Bugahyeon-ro, Seoul 03759, Republic of Korea. [5]Department of Chemical and Biomolecular Engineering, North Carolina State University, Raleigh, NC 27695, USA. [6]Material & Component Convergence R&D Department, Korea Institute of Industrial Technology, Ansan-si 15588, Republic of Korea. [7]These authors contributed equally: Ji-Hye Kim, Sooyoung Kim. ✉e-mail: jso@kitech.re.kr; hjkoo@seoultech.ac.kr

prevents the otherwise low viscosity metal from freely flowing. Spreading GaLMs onto most surfaces requires applying a force to break the oxide shell[28,29].

It is possible to remove the oxide shell using, for example, a strong acid or base. In the absence of the oxide, GaLMs beads up on nearly all surfaces due to their enormous surface tension, but there are exceptions; GaLMs wet metal substrates. Ga forms metallic bonds with other metals through a process called 'reactive wetting'[30–32]. Such reactive wetting is usually studied in the absence of surface oxides to promote metal-metal contact. However, even when GaLMs have native oxides, there are reports of metal-metal contacts forming when the oxide breaks at the contact on the smooth metallic surface[29]. Reactive wetting results in low contact angles and favorable wetting behavior with most metallic substrates[33–35].

To date, many studies have been conducted to utilize the favorable reactive wetting property of GaLMs with metals for patterning GaLMs. For example, GaLMs have been applied on patterned solid metal traces through spreading, rolling, jetting, or depositing with a shadow mask[34–38]. The selective wetting of GaLMs on solid metals enables GaLMs to form stable and well-defined patterns. Yet, the large surface tension of GaLMs makes it difficult to create thin films of uniform height even on metal substrates. To address this, Lacour et al. reported a method for producing a smooth, flat, thin film of GaLM over a large area by evaporating pure gallium on a gold-coated microstructured substrate[37,39]. This approach requires vacuum deposition, which is slow. Furthermore, GaLMs are often not allowed in such equipment due to possible embrittlement[40]. Evaporation also deposits material over the entire substrate and thus requires a stencil to provide patterning. We sought a way to create smooth, thin films and patterns of GaLMs by designing topographical metallic features that GaLMs would spontaneously and selectively wet in the absence of a native oxide. Herein, we report the spontaneous selective wetting of EGaIn (a representative GaLM) without the oxide by utilizing the unique wetting behavior on a lithographically structured metallic substrate. We created lithographically-defined surface structures in microscale to study imbibition and thereby control the wetting of oxide-free liquid metal. The enhanced wetting characteristics of EGaIn on a metallic surface with microstructure are explained through a numerical analysis based on the Wenzel's model and imbibition process. Finally, we demonstrate the large-area deposition and patterning of EGaIn via the imbibition-induced, spontaneous and selective wetting on a microstructured, metal-deposited surface. Stretchable electrodes and strain sensors comprising the EGaIn patterns are introduced as potential applications.

## Results

### Imbibition-induced wetting of EGaIn liquid metal

Imbibition is capillary-driven transport where liquid invades a textured surface favorable to the spreading liquid[41]. We investigated the wetting behavior of EGaIn on the metal-deposited, microstructured surface in HCl vapor (Fig. 1). Copper was chosen as the metal for the underlying surface. On flat copper surfaces, EGaIn showed a low contact angle of <20° in the presence of HCl vapor, due to reactive wetting[31] (Supplementary Fig. 1). We measured similar contact angles of EGaIn on bulk copper and a copper film sputter-coated on polydimethylsiloxane (PDMS).

To evaluate the role of topography on wetting, PDMS substrates with post and pyramid patterns were prepared, and copper was deposited on them with a titanium adhesion layer (Fig. 1a). It was confirmed that the microstructured surfaces of PDMS substrates were conformally coated with copper (Supplementary Fig. 2). The time-dependent contact angles of EGaIn on the patterned and flat copper-deposited PDMS (Cu/PDMS) are shown in Fig. 1b. The contact angles of EGaIn on the patterned Cu/PDMS decrease to 0° in ~1 min. The enhanced wetting of EGaIn by the microstructures can be explained by

the Wenzel equation $\cos\theta_{rough} = r\cos\theta_0$, where $\theta_{rough}$ denotes the contact angle on the rough surface, $r$ the surface roughness (= actual area/apparent area), and $\theta_0$ the contact angle on the flat surface. Since $r$-values of the post- and pyramid-patterned surfaces are 1.78 and 1.73, respectively, the results of the enhanced wetting of EGaIn on the patterned surfaces agree well with the Wenzel model. This also implies that a drop of EGaIn sitting on the patterned surface penetrates into the grooves of the underlying topography. Importantly, this results in a flat film with uniform height, contrary to the case of EGaIn on the non-patterned surface (Supplementary Fig. 1).

From Fig. 1c, d (Supplementary Movie 1), it can be observed that after 30 s, the apparent contact angle approaches 0° and EGaIn starts to spread further from the edge of the droplet, which is induced by imbibition (Supplementary Movie 2 and Supplementary Fig. 3). Prior studies on flat surfaces have attributed this time scale for reactive wetting as a transition from inertial wetting to viscous wetting[29]. The dimension of the topography is one of the key factors for determining whether imbibition occurs or not. Bico et al. thermodynamically derived the critical contact angle of imbibition, $\theta_c$, by comparing the surface energy before and after the imbibition[41] (see Supplementary Discussions for details). The resulting $\theta_c$ is defined as $\cos\theta_c = (1 - \phi_S)/(r - \phi_S)$, where $\phi_s$ denotes the fractional area of the top of the posts and $r$ represents the surface roughness. Imbibition can occur when $\theta_c > \theta_0$, i.e., the contact angle on a flat surface. For a post-patterned surface, $r$ and $\phi_s$ are calculated as $1 + \{(2\pi RH)/d^2\}$ and $\pi R^2/d^2$, respectively, where $R$ denotes the radius of a post, $H$ the height of a post, and $d$ the distance between the centers of the two posts (Fig. 1a). For the post-patterned surface in Fig. 1a, $\theta_c$ is 60°, which is larger than the $\theta_0$ (~25°) of oxide-free EGaIn on flat Cu/PDMS in HCl vapor. Consequently, the EGaIn droplet easily invades the post-patterned Cu-deposited surface in Fig. 1a through imbibition.

### Effect of dimensions of the post patterns on the imbibition wetting

To investigate the effect of the topographical dimensions of the patterns on the wetting and imbibition of EGaIn, we varied the dimensions of the Cu-coated posts. Figure 2 reports the contact angles and imbibition of EGaIn on these substrates. The distance between the posts, $l$, and the diameter of the posts, $D$, are equal and range from 25 to 200 μm. The height of 25 μm is constant for all the posts. $\theta_c$ decreases as the dimensions of the posts increase (Table 1), which implies that imbibition is less likely to occur on substrates with bigger posts. For all the dimensions tested, $\theta_c$ values are larger than $\theta_0$ and imbibition is expected to occur. However, imbibition is rarely observed for the post-patterned surface with $l$ and $D$ of 200 μm (Fig. 2e).

The other criterion to determine whether the imbibition of liquid could happen or not is the pinning of liquid on the post-patterned surface. Courbin et al. reported that the imbibition of liquid droplets on a post-patterned surface occurs when (1) the posts are sufficiently tall; (2) the distance between the posts is sufficiently small; and (3) the contact angle of the liquid on the surface is sufficiently small[42]. Numerically, the $\theta_0$ of the liquid on a flat surface comprising the same substrate material should be smaller than the critical contact angle for pinning, $\theta_{c,pin}$, for imbibition without pinning between posts, where $\theta_{c,pin} = \arctan(H/\{(\sqrt{2}-1)l\})$ (see Supplementary Discussions for details). The values for $\theta_{c,pin}$ depend on the post dimensions (Table 1). A dimensionless parameter, $L = l/H$, is defined for judging whether or not imbibition occurs. For imbibition, $L$ should be smaller than threshold criterion, $L_c = 1/\{(\sqrt{2}-1)\tan\theta_0\}$. For EGaIn on copper substrate ($\theta_0 = 25°$), $L_c$ is 5.2. Since $L$ is 8 for 200 μm posts, which is larger than the $L_c$ value, the imbibition of EGaIn does not occur. For further validation of the effect of geometry, we observed imbibition for various $H$ and $l$ (Supplementary Fig. 5 and Supplementary Table 1). The result showed good agreement with our calculation. Thus, $L$

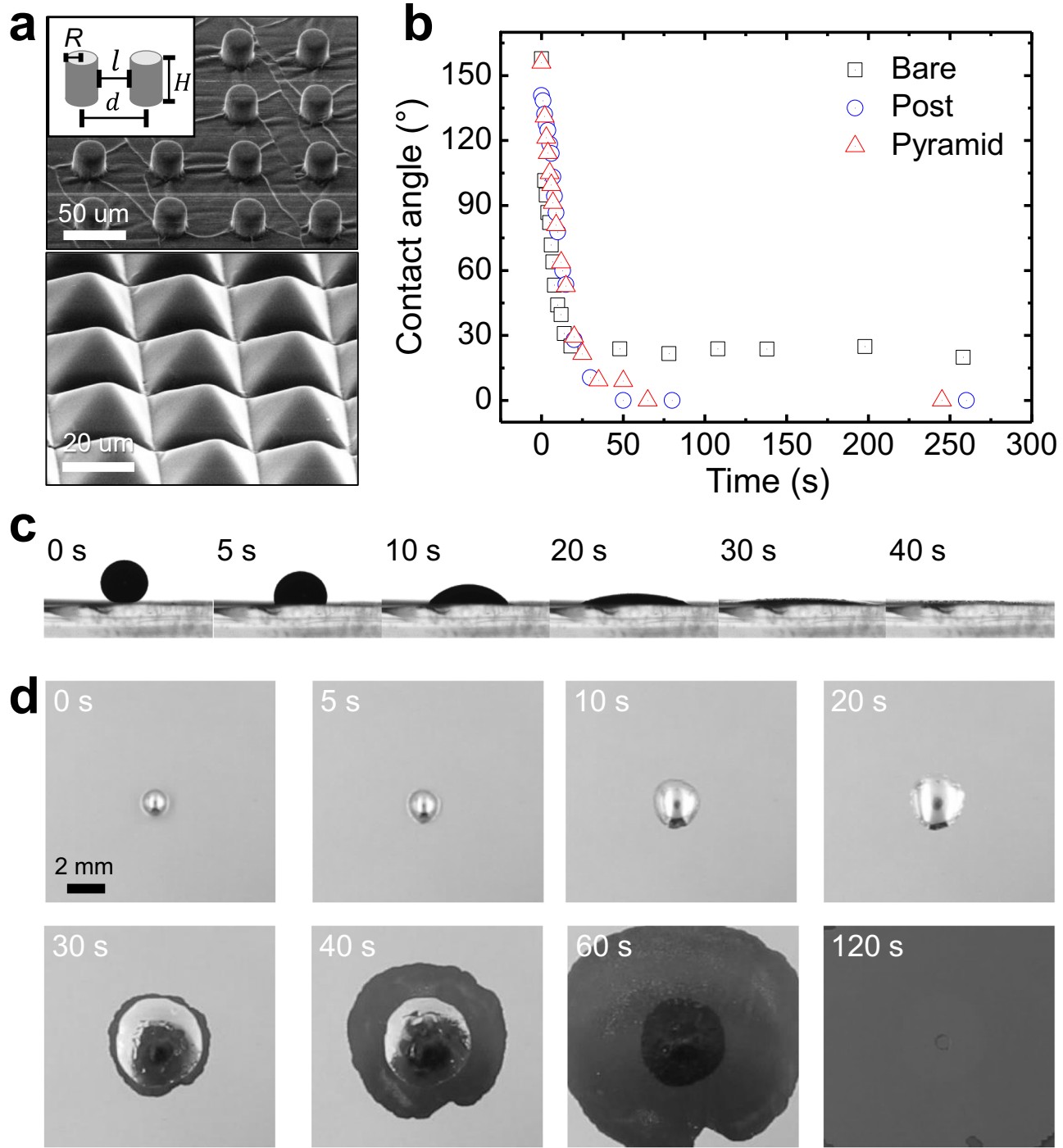

**Fig. 1 | Imbibition-induced wetting phenomena of EGaIn on a metal-deposited surface with microstructures in the presence of HCl vapor. a** Scanning electron microscope (SEM) images of posts ($D$ (diameter) = $l$ (spacing) = 25 μm, $d$ (distance between posts) = 50 μm, $H$ (height) = 25 μm) and pyramids (width = 25 μm, height = 18 μm)-shaped microstructures on Cu/PDMS substrates. **b** Time-dependent changes in contact angles on flat substrates (without microstructures), and arrays of posts and pyramids comprising PDMS coated with Cu. **c, d** Time-lapse (**c**) side- and (**d**) top-view images of EGaIn wetting on a surface with posts in the presence of HCl vapor.

appears to be an effective predictor of imbibition; liquid metal stops undergoing imbibition through the pinning when the distance between posts is relatively large compared to the height of posts.

The wetting property can be determined according to the surface composition of substrate. We investigated the effect of surface composition on the wetting and imbibition of EGaIn by co-depositing Si and Cu onto the posts and flat surfaces (Supplementary Fig. 6). As Cu content of the flat, Si/Cu binary surfaces increases from 0 to 75%, the contact angle of EGaIn decreases from ~160° to ~80°. For the surface of

75% Cu/25% Si, $\theta_0$ is ~80°, which corresponds to $L_c$ is 0.43 according to the above definition. Since $L$ for the posts with $l = H = 25$ μm is 1 which is larger than the threshold value $L_c$, imbibition does not occur on the post-patterned 75% Cu/25% Si surface due to pinning. As the contact angle of EGaIn increases with the addition of Si, higher $H$ or lower $l$ is required to overcome the pinning and for imbibition to occur. Thus, since the contact angle (i.e., $\theta_0$) depends on the chemical compositions of surfaces, it can also determine whether imbibition occurs or not on microstructures.

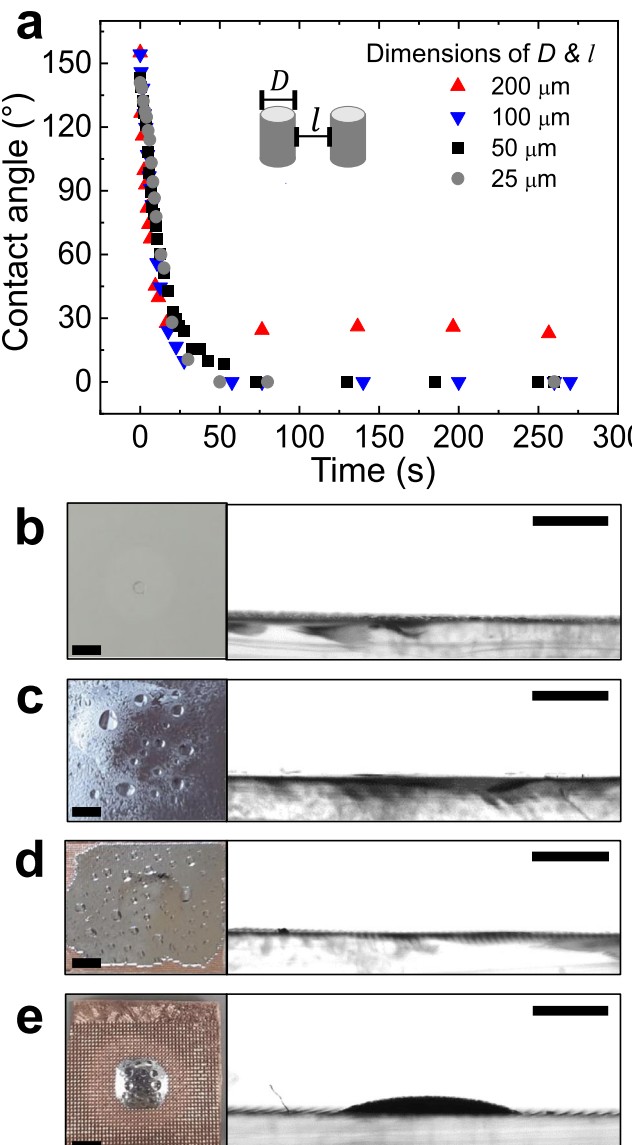

**Fig. 2 | Effect of the dimensions of post patterns on the contact angles and imbibition of a droplet of EGaIn. a** Time-dependent change in contact angles of EGaIn on the Cu/PDMS surfaces with various dimensions of posts after exposure to HCl vapor. **b**–**e** Top and side views of EGaIn wetting. **b** $D = l = 25\,\mu m$, $r = 1.78$. **c** $D = l = 50\,\mu m$, $r = 1.39$. **d** $D = l = 100\,\mu m$, $r = 1.20$. **e** $D = l = 200\,\mu m$, $r = 1.10$. The height of all the posts is $25\,\mu m$. The images were captured at least 15 min after the HCl vapor exposure. The liquid droplets on EGaIn are water generated through the reaction between gallium oxide and HCl vapor[47]. All scale bars in (**b**–**e**) are 2 mm.

The imbibition of EGaIn on the patterned Cu/PDMS can direct the wetting of the liquid metal into useful patterns. To assess the minimum number of post rows to induce imbibition, the wetting characteristics of EGaIn were observed on the Cu/PDMS with post-patterned lines comprising different numbers of post rows from 1 to 101 (Fig. 3). The wetting dominantly occurs along the post-patterned region. The imbibition of EGaIn is reliably observed, and the imbibition length increases with the number of the rows of posts. The imbibition barely occurs when two or fewer rows of posts are present. This may result from the increased capillary pressure. For imbibition to occur along post patterns, the capillary pressure by the curvature of the head of EGaIn should be overcome (Supplementary Fig. 7). With the assumption of the $12.5\,\mu m$ radius of curvature of the EGaIn head for a single row of post pattern, the capillary pressure is -0.98 atm (-740 Torr). Such a high Laplace pressure likely prevents EGaIn from imbibition-driven wetting.

**Table 1 | $\theta_c$, $\theta_{(c,pin)}$, and $L$ as a function of the dimensions of posts**

| $D$ and $l$ (µm) | $H$ (µm) | $\theta_O$ | $\theta_c$ | $\theta_{(c,pin)}$ | $L_c$ | $L(l/H)$ | Imbibition |
|---|---|---|---|---|---|---|---|
| 25 | 25 | ~25° | 60° | 68° | 5.2 | 1 | ○ |
| 50 | | | 48° | 50° | | 2 | ○ |
| 100 | | | 36° | 31° | | 4 | ○ |
| 200 | | | 27° | 17° | | 8 | × |

Moreover, fewer numbers of post rows might decrease the imbibition force, which is driven by capillary action between EGaIn and the posts.

### Patterning of liquid metal via imbibition-driven selective wetting

The imbibition of EGaIn on the post-patterned Cu/PDMS enables the EGaIn patterning via selective wetting (Fig. 4). When EGaIn droplets are placed on the post-patterned region and are exposed to HCl vapor, EGaIn droplets first collapse to form a small contact angle as the acid removes the oxide skin. Subsequently, imbibition starts from the edge of the droplets. The large-area patterning of EGaIn with a centimeter scale can be achieved (Fig. 4a, c). Since imbibition only occurs with a topographical surface, EGaIn wets only the post-patterned region and nearly stops wetting when reaching the flat surface. Consequently, the sharp boundaries of EGaIn patterns are observed (Fig. 4d, e). In Fig. 4b, it can be observed that EGaIn invades the non-patterned region, especially around the positions where the EGaIn droplets are initially placed. This occurs because the minimum diameter of EGaIn droplets used in this study exceeds the width of the patterned letters. The EGaIn droplets were placed on the patterned regions by manually injecting through a 27-G needle and a syringe, producing droplets with a minimum size of 1 mm. This issue could be overcome if smaller EGaIn droplets are used. Taken in sum, Fig. 4 shows that spontaneous EGaIn wetting can be induced and guided on the microstructured surface. Such a wetting process is relatively fast and does not require external force to achieve a complete wetting, compared to previous works (Supplementary Table 2).

### Electrical stability of imbibition-induced EGaIn thin film

Owing to the liquid characteristics of EGaIn, the EGaIn-coated Cu/PDMS (EGaIn/Cu/PDMS) can be used for flexible and stretchable electrodes. Figure 5a compares the change in the resistance of the pristine Cu/PDMS and EGaIn/Cu/PDMS under different strains. The resistance of the Cu/PDMS drastically increases with strain, whereas that of the EGaIn/Cu/PDMS maintains low resistance while strained. Figure 5b and d display the SEM images and corresponding EDS data of the pristine Cu/PDMS and EGaIn/Cu/PDMS before and after the strain application. For the pristine Cu/PDMS, the strain generates cracks in the rigid copper film sputtered on PDMS due to elastic mismatch. Conversely, for the EGaIn/Cu/PDMS, EGaIn remains well-covered on the Cu/PDMS substrate and maintains electrical continuity without any crack or significant deformation even after the strain application. The EDS data confirms that gallium and indium from EGaIn are uniformly spread out on the Cu/PDMS substrate. Notably, the thickness of the EGaIn film is uniform and comparable to the height of the post. This is also confirmed by further topographical analysis, where the relative difference between the thickness of the EGaIn film and the height of the post is <10% (Supplementary Fig. 8 and Table 3). This imbibition-based wetting could enable the thickness of EGaIn coating to be well-controlled and stably maintained over a large area which is otherwise challenging due to its fluid nature. Fig. 5c and e compare the stability of electrical conductivity of the pristine Cu/PDMS and EGaIn/Cu/PDMS against the strain. For the demonstration, an LED turns on while connected to the pristine Cu/PDMS or EGaIn/Cu/PDMS electrodes. When the pristine Cu/PDMS is stretched, the LED light turns off. The EGaIn/

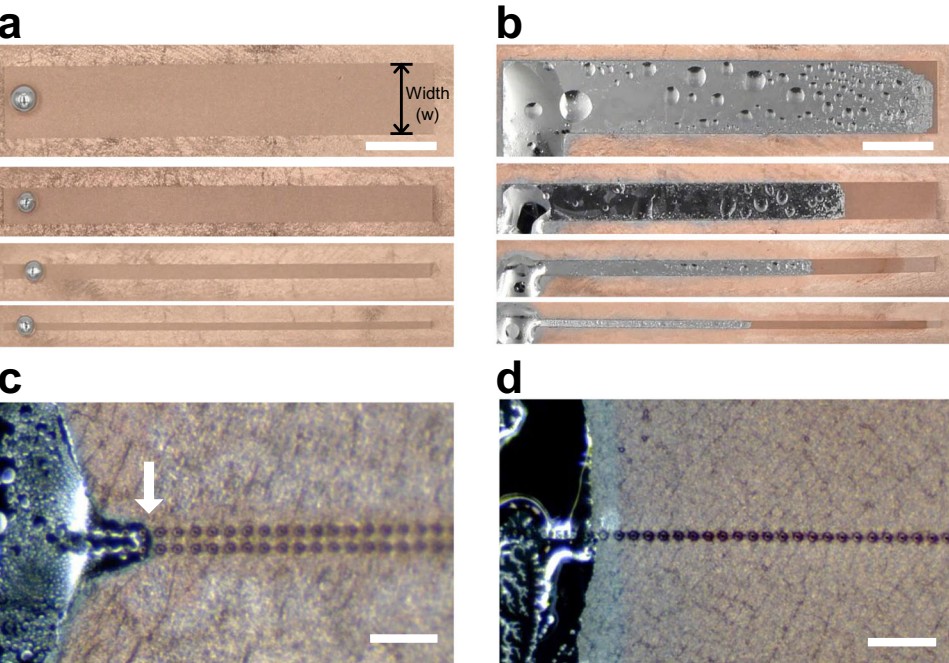

**Fig. 3 | Effect of the width of rows of posts of Cu/PDMS on the imbibition of EGaln. a** EGaln droplets sitting on the post-patterned Cu/PDMS with different widths of the patterns ($w$) in air (before exposure to HCl vapor). From the top, the numbers of the rows of posts are 101 ($w = 5025\,\mu m$), 51 ($w = 2525\,\mu m$), 21 ($w = 1025\,\mu m$), and 11 ($w = 525\,\mu m$). **b** Directional wetting of EGaln in (**a**) 10 min after HCl vapor exposure. **c, d** Wetting of EGaln on the Cu/PDMS with the (**c**) two rows ($w = 75\,\mu m$) and (**d**) one row ($w = 25\,\mu m$) of the post patterns. The images were captured 10 min after the HCl vapor exposure. The scale bars in (**a, b**) and (**c, d**) are 5 mm and 200 μm, respectively. The arrow in (**c**) indicates the curvature of the head of EGaln through imbibition.

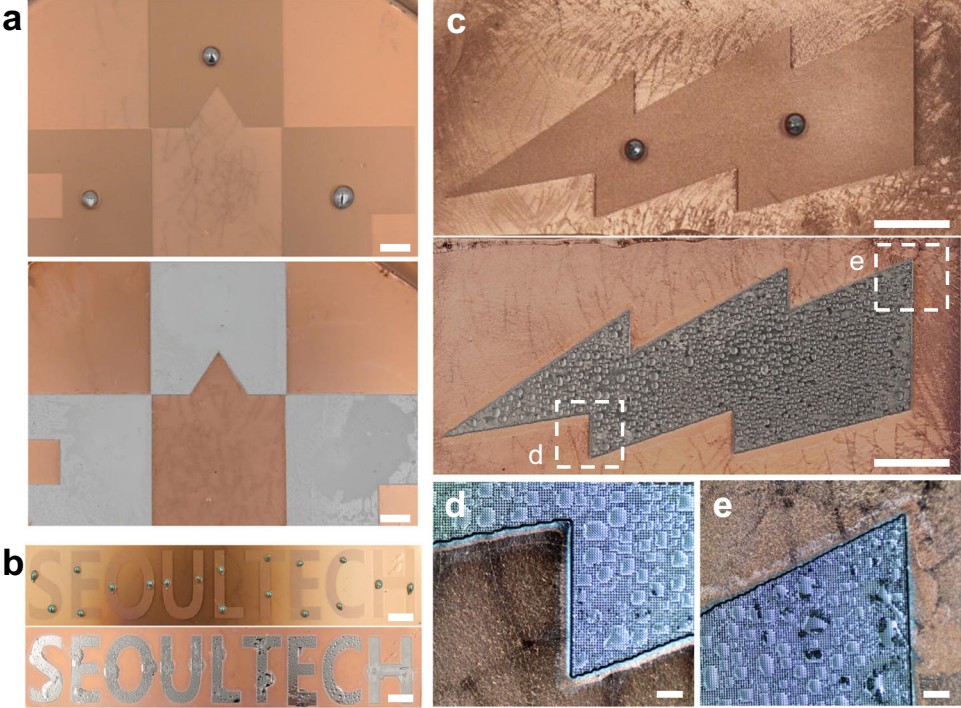

**Fig. 4 | EGaln patterning via imbibition-driven selective wetting. a** University emblem, **b** letters, and **c** lightning shape. Regions for imbibition are covered with the arrays of posts with $D = l = 25\,\mu m$. **d, e** Enlarged images of edges in (**c**). The scale bars in (**a–c**) and (**d, e**) are 5 mm and 500 μm, respectively. In (**c–e**), the small droplets on the surface after imbibition are water through the reaction between gallium oxide and HCl vapor. No significant effect of water formation on the wetting was observed. The water could be easily removed by a simple drying process.

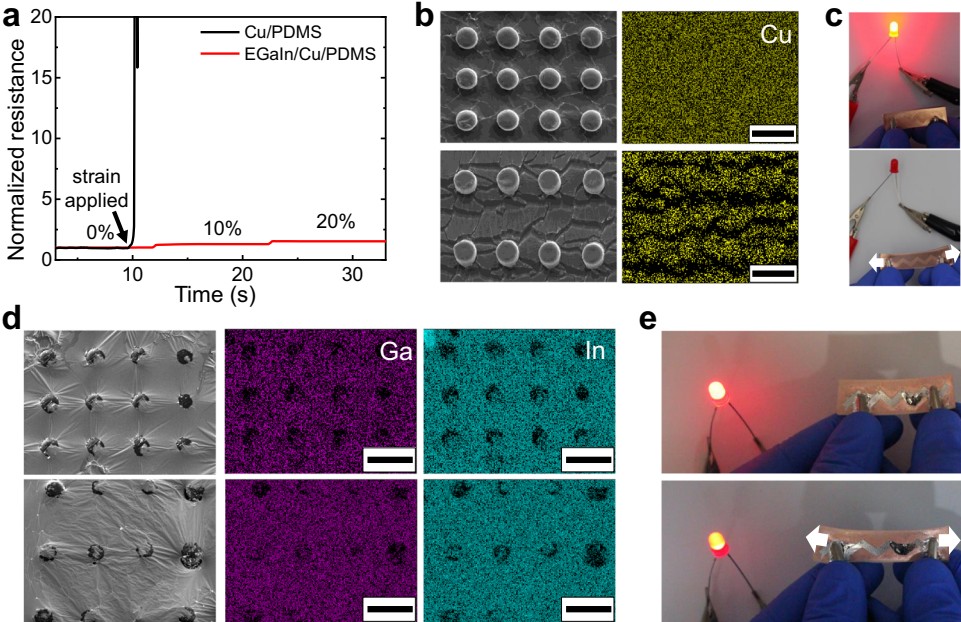

**Fig. 5 | Effect of strain on the electrical stability of pristine Cu/PDMS and EGaIn-coated Cu/PDMS (EGaIn/Cu/PDMS). a** Change in the normalized resistance with increasing strain on Cu/PDMS and EGaIn/Cu/PDMS. **b, d** SEM images and energy-dispersive X-ray spectroscopy (EDS) analysis before (top) and after (bottom) strain (50 - 70%) is loaded to (**b**) Cu/PDMS and (**d**) EGaIn/Cu/PDMS. **c, e** An LED connected with (**c**) Cu/PDMS and (**e**) EGaIn/Cu/PDMS before (top) and after (bottom) stretching (~30% strain). The scale bars in (**b**) and (**d**) are 50 μm.

Cu/PDMS electrode, however, maintains the electrical connection even under strain, and the LED light only dims slightly due to the increased resistance of the electrode.

Figure 6a shows the change in the resistance of the EGaIn/Cu/PDMS depending on strain from 0 to 70%. The resistance increases and recovers proportional to the strain, which agrees well with Pouillet's law ($R/R_O = (1 + \varepsilon)^2$) for incompressible materials, where $R$ is resistance, $R_O$ is the initial resistance and $\varepsilon$ is strain[43]. Other studies have proposed that, upon stretching, solid particles in liquid media may be rearranged and more uniformly distributed with better connectivity, mitigating the increase in resistance[43,44]. In this work, however, the conductor is >99% liquid metal by volume since the Cu films are only 100 nm thick. Thus, we do not anticipate Cu to contribute significantly to the electromechanical performance of the conductor.

Gauge factor (GF) denotes the sensitivity of the sensors, which is defined as the ratio of change in electric resistance to the change in strain[45]. The GF increases from 1.7 at 10% strain to 2.6 at 70% strain due to geometric changes to the metal. The GF values of the EGaIn/Cu/PDMS are modest relative to other strain sensors. As a sensor, although its GF may not be remarkably high, the EGaIn/Cu/PDMS exhibits reliable resistance change in response to strain with a low signal-to-noise ratio. To evaluate the conductivity stability of EGaIn/Cu/PDMS, the resistance was monitored during repeated stretching–releasing cycles with 30% strain. As shown in Fig. 6b, the resistance value is maintained after 4000 stretching cycles within 10% variation, which might be due to the continuous formation of the oxide skin during the repetitive stretching cycles[46]. Thus, it is confirmed that the EGaIn/Cu/PDMS possesses long-term electrical stability as a stretchable electrode as well as signal reliability as a strain sensor.

## Discussion
In this paper, we discuss the enhanced wetting characteristics of GaLMs on a microstructured metallic surface induced by imbibition. The spontaneous complete wetting of EGaIn was realized on the post- and pyramid-patterned metallic surfaces in the presence of HCl vapor. This can be numerically explained based on the Wenzel's model and imbibition process, which reveals the dimensions of the post

microstructures required for imbibition-induced wetting. The spontaneous and selective wetting of EGaIn directed by a microstructured metallic surface enables the large-area uniform coating and patterning of the liquid metal. The EGaIn-coated Cu/PDMS substrates maintain electrical connectivity even in a stretched state and after repetitive stretching cycles, which is confirmed using SEM, EDS, and resistance measurements. Moreover, the resistance of the EGaIn-coated Cu/PDMS reversibly and reliably changes proportional to the applied strain, which shows its potential application as a strain sensor. The possible advantages of the imbibition-induced wetting principle of the liquid metal are: (1) Coating and patterning of GaLM can be realized without external force; (2) The wetting of GaLM on the copper-coated, microstructured surface is thermodynamically favorable, which makes the resulting GaLM film stable even under deformation; (3) Varying the height of the copper-coated posts enables the formation of the GaLM thin film with controlled thickness. Furthermore, this approach reduces the amount of the GaLM required for thin film formation since the pillars occupy part of the film. For example, by introducing an array of posts with 200 μm diameter (with 25 μm spacing between them), the volume of the GaLM required for the film formation (~9 μm³/μm²) reduces by up to 64%, compared to that for the film without posts (25 μm³/μm²). In this case, however, it needs to be considered that the theoretical resistance estimated by the Pouillet's law also increases nine times. Taken in sum, the unique wetting characteristics of the liquid metal discussed herein suggest an effective way to introduce the liquid metal on various substrates for stretchable electronics and other emerging applications.

## Methods
### Preparation of microstructured metallic surfaces
PDMS substrate was prepared by mixing Sylgard 184 (Dow Corning, USA) base and curing agent in 10:1 and 15:1 ratios for the stretching test and then curing in an oven at 60 °C. Copper or silicon was deposited using a custom-made sputtering system on a silicon wafer (Si wafer, Namkang Hi-Tech Co., Ltd, Republic of Korea) and PDMS substrate with a 10-nm Ti adhesion layer. The photolithography process was adopted on the Si wafer for the post and pyramid structures, and their patterns were transferred on PDMS substrates. The width and height of

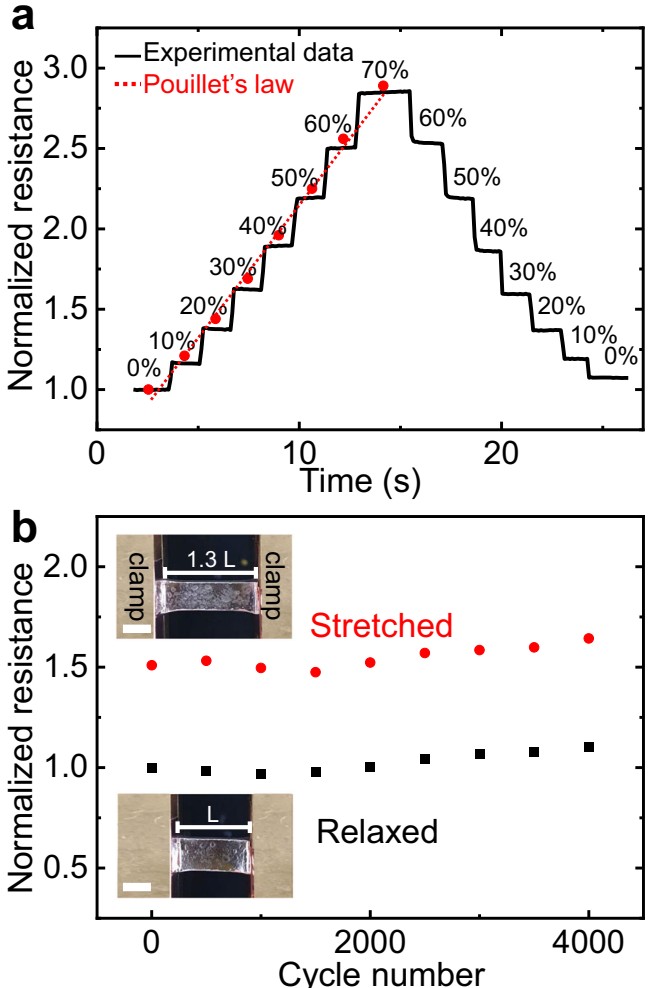

**Fig. 6 | Strain-dependent resistance of EGaIn/Cu/PDMS. a** Normalized resistance change of EGaIn/Cu/PDMS depending on strain in the range of 0–70%. The maximum strain reached before the PDMS breaks is 70% (Supplementary Fig. 9). The red dots are the theoretical values predicted from Pouillet's law. **b** The conductivity stability test of EGaIn/Cu/PDMS during repetitive stretching–releasing cycles. The cycle test utilized 30% strain. The scale bars in inset images are 0.5 cm. *L* is the initial length of EGaIn/Cu/PDMS before stretching.

the pyramid pattern were 25 and 18 μm, respectively. The post pattern had fixed height of 25 μm, 10 μm, and 1 μm, and its diameter and spacing varied from 25 to 200 μm.

## Contact angle measurements
The contact angle of EGaIn (gallium 75.5%/indium 24.5%, >99.99%, Sigma Aldrich, Republic of Korea) was measured using a drop-shape analyzer (DSA100S, KRUSS, Germany). The substrates were placed in a 5 cm × 5 cm × 5 cm glass chamber, and 4–5 μL of EGaIn droplets were placed on the substrates through a 0.5-mm diameter syringe. To form the HCl vapor environment, a 20-μL drop of HCl solution (37 wt.%, Samchun Chemicals, Republic of Korea) was placed next to the substrate, which was evaporated enough to fill the chamber within 10 s.

## Surface analysis
The surface was imaged with SEM (Tescan Vega 3, Tescan Korea, Republic of Korea). EDS (Tescan Vega 3, Tescan Korea, Republic of Korea) was performed for investigation of the elemental qualitative analysis and distribution. The surface topographies of EGaIn/Cu/PDMS were analyzed using an optical profilometer (The Profilm3D, Filmetrics, USA).

## Electrical conductivity measurement of patterned EGaIn under strain
To investigate the change in electrical conductivity during stretching cycles, the samples with and without EGaIn were clamped on the stretching equipment (Bending & Stretchable Machine System, SnM, Republic of Korea) and were electrically connected to a Keithley 2400 source meter. The resistance change according to the range of 0% to 70% strain applied to the sample was measured. For the stability test, the resistance change was measured during the 4000 cycles of 30% strain.

## Reporting summary
Further information on research design is available in the Nature Research Reporting Summary linked to this article.

## Data availability
The data that support the findings of this study are provided in the Supplementary information and Source Data file. Source data are provided with this paper.

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

## Acknowledgements

We thank Professor Hyunsik Yoon for his helpful comments on imbibition study. We gratefully acknowledge the support of this work provided by Basic Science Research Programs through the National Research Foundation of Korea (NRF) funded by the Ministry of Science and ICT (NRF-2021R1H1A1014396) and by the Ministry of Education (NRF-2021R1A6A1A03039981). This work has also been partially supported by Nano-Material Technology Development Program (No. 2017M3A7B8061942) through the National Research Foundation of Korea funded by the Ministry of Science and ICT.

## Author contributions

J.-H.K., S.K., J.-H.S. and H.-J.K. conceptualized and designed and the research; J.-H.K., and S.K. conducted the experiments and wrote the original draft; H.K. and S.W. helped the contact angle measurement; J.C. fabricated the metal-deposited substrates; M.D.D. reviewed and edited the manuscript; J.-H.S., and H.-J.K. supervised the work and reviewed and edited the manuscript.

## Competing interests

The authors declare no competing interests.
