## [Peer Review File · Nature Communications]

Imbibition-induced selective wetting of liquid metalREVIEWER COMMENTS

Reviewer #1 (Remarks to the Author):

Review of Imbibition-induced selective wetting of liquid metal

The primary claim of this manuscript is the introduction of a process that enables the patterning of liquid metal due to selective wetting through surface texturing. My overall impression is that this manuscript is tidy and reads well, however, I believe that much of this work has been shown before. The manuscript will still make a nice contribution to the literature, but not at the impact level that I perceive for Nature Communications.

End of the first paragraph: is 11C the melting point of Galinstan? I believe it is commonly known to be -19C.

"The formation of a native gallium oxide skin on the surface of GaLMs under ambient conditions reduces the surface tension, thereby enabling GaLMs to be stabilized in non-spherical shapes." I think this should say increases?

"...the oxide skin, which is a solid, mechanically inhibits the liquid metal from wetting and spreading onto surfaces unless a force enough to break the solid skin is applied. Here, we show that properly designing the topography of metal-coated surfaces can enable GaLMs to form thin films and patterns on surfaces, spontaneously, without an external force." I believe these statements overlook prior works on reactive wetting, which the authors refer to later in the manuscript. But these rang as incorrect to me. The work presented in this manuscript seems very similar to a combination of prior published works. Namely, the following come to mind:

1. Ozutemiz, Kadri Bugra, et al. "EGaIn–metal interfacing for liquid metal circuitry and microelectronics integration." *Advanced Materials Interfaces* 5.10 (2018): 1701596.
2. Kramer, Rebecca K., Carmel Majidi, and Robert J. Wood. "Masked deposition of gallium-indium alloys for liquid-embedded elastomer conductors." *Advanced functional materials* 23.42 (2013): 5292-5296.
3. Kramer, Rebecca K., et al. "Effect of microtextured surface topography on the wetting behavior of eutectic gallium–indium alloys." *Langmuir* 30.2 (2014): 533-539.
4. Doudrick, Kyle, et al. "Different shades of oxide: From nanoscale wetting mechanisms to contact printing of gallium-based liquid metals." *Langmuir* 30.23 (2014): 6867-6877.

The first two papers listed here are on selective wetting of liquid metals on solid metallic films and are cited in the current manuscript. The second two deal with the wetting behaviors of liquid metal, which seem foundational to the current manuscript, but these are not cited. It seems that authors are claiming a first understanding of the wetting behaviors of liquid metals. However, I believe the current manuscript supports these prior papers from 2014 but is not the first to use and understand this approach.

"The effect of surface topography and its geometry on the wetting of EGaIn was investigated." Similar to the comments above, this is nearly the same claim as was put forward in "Effect of microtextured surface topography on the wetting behavior of eutectic gallium–indium alloys." *Langmuir* 30.2 (2014): 533-539.

"From Figs. 1c and d, it can be observed that after 30 s, the apparent contact angle approaches theta and EGaIn starts to spread further from the edge of the droplet; this phenomenon is called imbibition." What is significant about 30 s? This time stamp and observation that the wetting behavior changes would seem to agree with Figure 4C of the *Langmuir* paper referenced above, which shows two distinct time scales of wetting: first inertial wetting, and then viscous wetting.

"Figure 2 reports the contact angles and imbibition of EGaIn on these substrates." Which contact angle? A recent paper looked at the utility of different contact angle measurements.

Joshiyura, Ishan D., et al. "Are Contact Angle Measurements Useful for Oxide-Coated Liquid Metals?" *Langmuir* 37.37 (2021): 10914-10923.

This paper is cited in an odd place in the current manuscript, and the results from the recent paper do not seem properly utilized in the current work.

"The distance between the posts of 200 um is sufficiently large that the liquid metal stops undergoing imbibition through pinning." What does "sufficiently large" mean here? "Sufficiently" is used multiple times in this paragraph, and I wonder how to generalize this result. Can the authors determine a quantitative threshold and introduce a generalized relationship between post geometry and wetting that others can use?

"The gauge factor of the EGaIn/Cu/PDMS is 2.6 at 70% strain, which again indicates that the resistance is less sensitive to strain or deformation than other strain sensors reported." This result would perhaps agree with biphasic liquid metal formulations in the literature, which have shown that including conductive solids in liquid metals suppress the electrical dependence to strain. For example, see:

1. Guo, Rui, et al. "Ni-GaIn Amalgams Enabled Rapid and Customizable Fabrication of Wearable and Wireless Healthcare Electronics." *Advanced Engineering Materials* 20.10 (2018): 1800054.
 2. Daalkhaijav, Uranbileg, et al. "Rheological modification of liquid metal for additive manufacturing of stretchable electronics." *Advanced materials technologies* 3.4 (2018): 1700351.
 3. Liu, Shanliangzi, Dylan S. Shah, and Rebecca Kramer-Bottiglio. "Highly stretchable multilayer electronic circuits using biphasic gallium-indium." *Nature Materials* 20.6 (2021): 851-858.
- Maybe the solid metal film is cracking and integrating into the liquid metal to produce a similar effect?

The Wenzel model is mentioned several times in this manuscript, but I'd like to see more analysis. Prior works have found that certain surface textures produce a Cassie state and prohibit the wetting behavior of liquid metal. This current manuscript would seem to show a confluence of reactive wetting and Cassie-Baxter to Wenzel transition. I think this manuscript would be much improved if the authors can identify transitions between Cassie and Wenzel states, and discuss them within the context of reactive wetting, which is present due to the metallic film but not considered by classical wetting models.

"This can be numerically explained based on Wenzel's model and imbibition process, which reveals the dimensions of the post microstructures required for imbibition-induced wetting." Similar to the comment above, I do not believe this statement is met by the manuscript.

There are three advantages to imbibition-induced wetting given at the end of the manuscript. The first two are already shown in prior works, although the third seems like a novel claim: "Varying the height of the copper-coated posts potentially enables the formation of the GaLM thin film with controlled thickness. Furthermore, this approach reduces the amount of the GaLM required for thin film formation since the pillars occupy part of the film." The second sentence here requires more quantification. Can the authors provide a strain model? This again brings me back to recent works on solid-liquid biphasic formulations and their impacts on the strain sensitivity of liquid metals. As this is a more structured geometry than those prior works, it would be nice if the authors could relate their electromechanical measurements to the prior proposed models.

Overall, I enjoyed reading this manuscript and I commend the authors for the presentation of their results. I believe this manuscript will make a nice contribution to the literature, though I also believe it is lacking critical analysis and is missing the opportunity to provide new generalized relationships between surface composition, texture, and liquid metal wetting behaviors.

Reviewer #2 (Remarks to the Author):

This work reports the spontaneous selective wetting phenomena of liquid metal on a micro-structured copper substrate in the presence of HCL vapor. And effects of the microstructures to enhance the EGaIn wetting were tested. The article can be of active reference for the society. It is recommended for acceptance after further improvement.

Some specific comments:

1. The originality of the method can be further improved especially compared with those existing similar endeavors.

2. The liquid droplets on EGaIn are water. The authors may want to further explain whether there is any impact when used, and how to remove the water.

3. In this work, the article used normalized resistance. What's the resistance values of EGaIn/Cu before stretched and after relaxed? Compared them with that of Cu or EGaIn.

4. Besides, whether the 10% variation of resistance value after 4000 stretching cycles is due to the cracks in the rigid copper film? Please give some explanation.

Reviewer #3 (Remarks to the Author):

The manuscript describes the imbibition-induced, spontaneous and selective wetting of EGaIn on metallized surfaces with microscale topographic features. The manuscript is well written and contains noteworthy results for the flexible electronics community.

The following are a few comments to the authors:

Line 119: "this phenomenon is called imbibition": The authors could add a reference here.

Line 157: The equation could be misread and this reviewer suggests using a second bracket for all of the denominator.

Line 163: "Effect of the width of posts": Width of posts may be misleading here, because it could mean the diameter of the posts rather than the overall width of rows of posts. The authors may consider rephrasing.

Line 234 and Figure 6: Looking at Figure 6, it appears that the gauge factor is strain dependent over the tested strain range with the gauge factor increasing with increasing strain. If this is true, what is the reason for this? Also, should the gauge factor be defined in the manuscript?

Line 249, Discussion: The discussion mentions the testing of pyramid-patterned metallic surfaces, even though the manuscript largely focuses on post-patterned metallic surfaces (besides Fig. 1a). The authors should either limit the discussion to the post-patterned structures or include results on the pyramid-patterned surfaces as well.

The manuscript nicely highlights the upper limits for the post geometry. What about the other end of the spectrum; are there any limitations with respect to the smallest post geometries that could be considered?

The authors should discuss if there are any requirements with respect to the conformity of the metal film on the 3D PDMS patterns? As an example, how well were the sidewalls of the PDMS coated with metal and how critical is that for the imbibition-induced wetting process?

REVIEWER COMMENTS

Reviewer #1 (Remarks to the Author):

Review of Imbibition-induced selective wetting of liquid metal

The primary claim of this manuscript is the introduction of a process that enables the patterning of liquid metal due to selective wetting through surface texturing. My overall impression is that this manuscript is tidy and reads well, however, I believe that much of this work has been shown before. The manuscript will still make a nice contribution to the literature, but not at the impact level that I perceive for Nature Communications.

Response: We appreciate the reviewer's comment. We did our best to answer the questions and comments as follows herein. We hope that our response better highlights the originality and impact of this work. Briefly, we distinguish two types of wetting behaviors reported in the literature to give context.

- (1)** *Liquid metal with native oxide.* Several studies have shown that the oxide promotes adhesion to nearly any smooth surface, but not to surfaces with nano-scale roughness. The oxide can be broken (and reform) to help promote better contact to substrates. The oxide also prevents metal from flowing freely. We keep this discussion brief since in our studies, we have kept the metal oxide-free throughout by using acid.
- (2)** *Liquid metal without the native oxide.* In the absence of the oxide, liquid metals only wet to other metals via reactive wetting, but in a way that lacks control. Here, we use sub-mm structures (arrays of posts) to "imbibe" the metal and thereby control the direction and shape of the wetting. The exciting aspect of this approach is that a droplet of liquid metal can be placed in one location (on a surface coated with a thin layer of solid metal), and then directionally wet based on the spacing and pattern of posts. This approach also results in structures of liquid metal with controlled height. Finally, although it was not the goal of this work, we were surprised to find that the structures produce electrical conductors that do not exhibit a significant dependence of resistance on strain, which is desirable for stretchable electronics.

End of the first paragraph: is 11C the melting point of Galinstan? I believe it is commonly known to be -19C.

Response: The recent review paper (Tang, S.-Y. et al. Gallium Liquid Metal: The Devil's Elixir, *Annu. Rev. Mater. Res.* **51**, 381 (2021)) reported that, even though $-19\text{ }^{\circ}\text{C}$ has been often reported as the melting point of Galinstan, the original source of that value has not been found in the literature. The actual melting point of Galinstan is about $10\text{ }^{\circ}\text{C}$, while the supercooling effect may cause the freezing point to drop down to $-19\text{ }^{\circ}\text{C}$.

"The formation of a native gallium oxide skin on the surface of GaLMs under ambient conditions reduces the surface tension, thereby enabling GaLMs to be stabilized in non-spherical shapes." I think this should say increases?

Response: Thank you for the suggestion. This sentence introduced unnecessary complexity since, in reality, the oxide forms two new interfaces (metal-oxide, oxide-ambient) while also providing a mechanical shell. To simplify the sentence while making the main point, we have changed it as follows.

(Page 3)

The formation of a solid native gallium oxide skin on the surface of GaLMs under ambient conditions provides a shell that stabilizes GaLMs in non-spherical shapes.

"...the oxide skin, which is a solid, mechanically inhibits the liquid metal from wetting and spreading onto surfaces unless a force enough to break the solid skin is applied. Here, we show that properly designing the topography of metal-coated surfaces can enable GaLMs to form thin films and patterns on surfaces, spontaneously, without an external force." I believe these statements overlook prior works on reactive wetting, which the authors refer to later in the manuscript. But these rang as incorrect to me. The work presented in this manuscript seems very similar to a combination of prior published works. Namely, the following come to mind:

1. Ozutemiz, Kadri Bugra, et al. "EGaIn-metal interfacing for liquid metal circuitry and microelectronics integration." *Advanced Materials Interfaces* 5.10 (2018): 1701596.
2. Kramer, Rebecca K., Carmel Majidi, and Robert J. Wood. "Masked deposition of gallium-indium alloys for liquid-embedded elastomer conductors." *Advanced functional materials* 23.42 (2013): 5292-5296.
3. Kramer, Rebecca K., et al. "Effect of microtextured surface topography on the wetting behavior of eutectic gallium-indium alloys." *Langmuir* 30.2 (2014): 533-539.
4. Doudrick, Kyle, et al. "Different shades of oxide: From nanoscale wetting mechanisms to contact printing of gallium-based liquid metals." *Langmuir* 30.23 (2014): 6867-6877.

The first two papers listed here are on selective wetting of liquid metals on solid metallic films and are cited in the current manuscript. The second two deal with the wetting behaviors of liquid metal, which seem foundational to the current manuscript, but these are not cited. It seems that authors are claiming a first understanding of the wetting behaviors of liquid metals. However, I believe the current manuscript supports these prior papers from 2014 but is not the first to use and understand this approach.

Response: We are thankful to the reviewer for giving us this opportunity to clarify the novelty and improve the paper.

This work: Simply stated, the key novelty of this paper is the use of “imbibition” of liquid metal to control the direction and extent of wetting of liquid metal without oxide. This patterning method is enabled by using microscale posts with various spacing. The posts are covered with a thin layer of solid-metal to promote “reactive wetting”, but we note that despite the substrate being coated everywhere with such a film, the liquid metal wets preferentially in the imbibition zone. There are secondary benefits of the work, including the ability to control film height and the ability to make elastic conductors with electromechanical behavior.

Prior work: We broadly classify prior studies in two different categories. The first category contains studies in which the GaLM features a surface oxide. Most of the references (papers 2, 3 and 4) listed by the reviewer fall into that category. The second category contains studies in which the GaLM does not feature an oxide layer.

In the first category, LM adheres to all surfaces except those that are rough (e.g. paper 4). In the second category, the bare LM does not wet any surface except metals, which occurs through well-known “reactive wetting”. The first category can overlap with the second in cases in which the oxide breaks and the metal gets exposed to the substrate (e.g. this occurs in papers 2, 3 and 4).

- Papers 1 and 2 are related to reactive wetting of GaLM on metal surfaces. As the reviewer correctly notes, reactive wetting between liquid and solid metals is very well known in the literature. We cite these papers and others in the introduction.

- Paper 4 reports the different adhesion behaviors of liquid metal on the micro-structured surface depending on rupturing of the original oxide shell. This paper is not closely related to our work, but still worth being cited.

- Paper 3 focused on the reactive wetting of GaLMs "with oxide skin" on the metallic surfaces of In and Sn. We apologize for missing it since it focuses on the metal with oxide (whereas our system lacks oxide). The reactive wetting was avoided by forming nanoscale roughness on the surface, which was explained based on the GaLMs at Cassie-Baxter regime. In contrast, our work uses larger topography (lithographic posts) to promote “imbibition” wetting and does so without the oxide present. Thus, we are able to control the shape and direction of wetting of bare liquid metal.

The points of emphasis / novelty of the present work come from: 1) large-area uniform flat coating of the LM via the imbibition-induced spontaneous wetting, 2) the ability to direct the wetting of the LM by using patterned microscale surface topography, 3) establishment of numerical conditions of underlying surface chemistry and topography required for the imbibition-induced wetting, and 4) practical demonstration examples of the resulting GaLM patterning including interesting electromechanical behavior.

We cited the paper 3 and 4 as follows. Also, we have gone through the manuscript and revised it to avoid confusion.

(Page 3)

This property has allowed GaLMs to be printed, injected into microchannels, and patterned by taking advantage of the interfacial stability enabled by the oxide^{19,22-27}. The solid oxide shell also allows GaLMs to adhere to most smooth surfaces, but prevents the otherwise low viscosity metal from freely flowing. Spreading GaLMs onto most surfaces requires applying a force to break the oxide shell^{28, 29}.

"The effect of surface topography and its geometry on the wetting of EGaIn was investigated." Similar to the comments above, this is nearly the same claim as was put forward in "Effect of microtextured surface topography on the wetting behavior of eutectic gallium–indium alloys." Langmuir 30.2 (2014): 533-539.

Response: We realize that sentence was too general and have changed it. The Langmuir paper elucidates how the reactive wetting of GaLMs “with oxides” can be “restrained” by forming random nano-roughness on the metallic surface, which is explained by “Cassie-Baxter model”. In our work, the metal “lacks oxide” due to the presence of acid. The wetting is “enhanced” by introducing well-defined microstructures coated with metal, which is explained by “Wenzel model and imbibition process”. We made a figure for direct comparison between the two works, which is shown below.

	Kramer et al., Langmuir, 2018	This work
		Oxide	○	×
Wetting regime	Cass-Baxter	Wenzel
Effect of surface topography	Providing nonwetting surface to EGaIn	Enhancing wetting of EGaIn
Type of roughness	Random roughness with scale less than tens of μm	Lithographically-controlled structures with scale of tens~hundreds of μm

Fig. R1. (Review only) A direct comparison of the conditions mainly focused on in Kramer et al.'s work and ours.

Thus, the Langmuir paper is relevant to our work in terms of reactive wetting of GaLMs on metallic surface, while the purpose of using topography on metallic surface and the underlying mechanisms are different (as is the use of bare liquid metal versus liquid metal with oxide). We changed the sentence to “We created lithographically-defined surface structures in microscale to study imbibition and thereby control the wetting of oxide-free liquid metal.” and revised the manuscript throughout to present the conditions of our study more clearly.

"From Figs. 1c and d, it can be observed that after 30 s, the apparent contact angle approaches theta and EGaIn starts to spread further from the edge of the droplet; this phenomenon is called imbibition." What is significant about 30 s? This time stamp and observation that the wetting behavior changes would seem to agree with Figure 4C of the Langmuir paper referenced above, which shows two distinct time scales of wetting: first inertial wetting, and then viscous wetting.

Response: We really appreciate the reviewer's helpful comment. We found that the interpretation of the wetting behavior of EGaIn on In foil (non-sputtered) in the Langmuir paper may be applicable to our system, and therefore, we have revised the corresponding sentence. In our case, it seems plausible that the metal first wets into the posts upon which it rests (first 30 s) and then starts to spread outward along with imbibition.

(Page 6)

From Figs. 1c and d (Supplementary Movie 1), it can be observed that after 30 s, the apparent contact angle approaches 0° and EGaIn starts to spread further from the edge of the droplet, which is induced by imbibition (Supplementary Movie 2 and Supplementary Fig. 3). Prior studies on flat surfaces have attributed this time scale for reactive wetting as a transition from inertial wetting to viscous wetting²⁹.

"Figure 2 reports the contact angles and imbibition of EGaIn on these substrates." Which contact angle? A recent paper looked at the utility of different contact angle measurements.

Response: The contact angle in figure 2 indicated the time-dependent change in “apparent” contact angles of EGaIn after HCl vapor treatment. We revised the caption to be more detailed.

(Page 7)

a. Time-dependent change in contact angles of EGaIn on the Cu/PDMS surfaces with various dimensions of posts after exposure to HCl vapor.

Joshiyura, Ishan D., et al. "Are Contact Angle Measurements Useful for Oxide-Coated Liquid Metals?." *Langmuir* 37.37 (2021): 10914-10923.

This paper is cited in an odd place in the current manuscript, and the results from the recent paper do not seem properly utilized in the current work.

Response: The paper by Joshiyura focuses on the role of the oxide on wetting behavior and shows how advancing angles, rather than static contact angles, should be used since it mechanically inhibits wetting. Thus, we thought it appropriate to

include it at the current place, but the sentence has been revised as follows.

(Page 3)

The solid oxide shell also allows GaLMs to adhere to most smooth surfaces, but prevents the otherwise low viscosity metal from freely flowing. Spreading GaLMs onto most surfaces requires applying a force to break the oxide shell^{28, 29}. It is possible to remove the oxide shell using, for example, a strong acid or base.

"The distance between the posts of 200 μm is sufficiently large that the liquid metal stops undergoing imbibition through pinning." What does "sufficiently large" mean here? "Sufficiently" is used multiple times in this paragraph, and I wonder how to generalize this result. Can the authors determine a quantitative threshold and introduce a generalized relationship between post geometry and wetting that others can use?

Response: We appreciate the reviewer's helpful comment. We defined and added a dimensionless parameter, $L = l/H$, to provide a generalized criterion of post geometry for imbibition. Rather than the distance (l) between the posts itself, the ratio of the height of the posts (H) and the distance (l) between the post is critical for imbibition to occur. To prove the validity of the new parameter, L , we performed further experiments with more dimensions including varied height, H . (H was previously fixed at 25 μm .) The result shows that it is possible to determine whether imbibition occurs or not based on L and L_c (i.e. its threshold criterion). We added more data and discussion to the manuscript and revised accordingly, as below.

(Page 8)

A dimensionless parameter, $L = l/H$, is defined for judging whether or not imbibition occurs. For imbibition, L should be smaller than threshold criterion, $L_c = 1/\{(\sqrt{2} - 1) \tan \theta_{eq}\}$. For EGaIn on copper substrate ($\theta_0 = 25^\circ$), L_c is 5.2. Since L is 8 for 200 μm posts, which is larger than the L_c value, the imbibition of EGaIn does not occur. For further validation of the effect of geometry, we observed imbibition for various H and l (Supplementary Fig. 5 and Supplementary Table 1). The result showed good agreement with our calculation. Thus, L appears to be an effective predictor of imbibition; liquid metal stops undergoing imbibition through the pinning when the distance between posts is relatively large compared to the height of posts.

Table 1 θ_c , $\theta_{c,pin}$, and L as a function of the dimensions of posts

D and l (μm)	H (μm)	θ_0	θ_c	$\theta_{c,pin}$	L_c	L (l/H)	Imbibition
25	25	$\sim 25^\circ$	60°	68°	5.2	1	○
50			48°	50°		2	○
100			36°	31°		4	○
200			27°	17°		8	×

(Supplementary data)

Supplementary Fig. 5 | Imbibition-induced wetting behavior of EGaIn depending on the dimensions of posts.

Supplementary Table 1. θ_c , $\theta_{c,pin}$, L_c and L as a function of the dimensions of posts.

D (μm)	l (μm)	H (μm)	θ_0	θ_c	$\theta_{c,pin}$	L_c	L (l/H)	Imbibition
200	25	25		57°	68°		1	○
25	25	10		44°	44°		2.5	○
25	200	25	~25°	16°	17°	5.2	8	×
25	25	1		16°	5°		25	×

"The gauge factor of the EGaIn/Cu/PDMS is 2.6 at 70% strain, which again indicates that the resistance is less sensitive to strain or deformation than other strain sensors reported." This result would perhaps agree with biphasic liquid metal formulations in the literature, which have shown that including conductive solids in liquid metals suppress the electrical dependence to strain. For example, see:

1. Guo, Rui, et al. "Ni-GaIn Amalgams Enabled Rapid and Customizable Fabrication of Wearable and Wireless Healthcare Electronics." *Advanced Engineering Materials* 20.10 (2018): 1800054.
2. Daalkhajav, Uranbileg, et al. "Rheological modification of liquid metal for additive manufacturing of stretchable electronics." *Advanced materials technologies* 3.4 (2018): 1700351.
3. Liu, Shanliangzi, Dylan S. Shah, and Rebecca Kramer-Bottiglio. "Highly stretchable multilayer electronic circuits using biphasic gallium-indium." *Nature Materials* 20.6 (2021): 851-858.

Maybe the solid metal film is cracking and integrating into the liquid metal to produce a similar effect?

Response: We appreciate the reviewer's helpful comments and suggestion. Although it was not our goal to make a strain invariant conductor, we were excited to see these results.

We have explored the possibility that the biphasic liquid metal formation affects the strain-dependent conductivity. In our system, the volume of liquid metal is more than 99%, which is much larger than that of Cu. Therefore, the effect of rearrangement of solid metal fragments from the Cu thin film in the biphasic portion on the gauge factor seems likely to be negligible. Based on the comment, we included this possibility in the text.

(Page 13)

Other studies have proposed that, upon stretching, solid particles in liquid media may be rearranged and more uniformly distributed with better connectivity, mitigating the increase in resistance^{44, 46}. In this work, however, the conductor is more than 99% liquid metal by volume since the Cu films are only 100 nm thick. Thus, we do not anticipate Cu to contribute significantly to the electromechanical performance of the conductor.

The Wenzel model is mentioned several times in this manuscript, but I'd like to see more analysis. Prior works have found that certain surface textures produce a Cassie state and prohibit the wetting behavior of liquid metal. This current manuscript would seem to show a confluence of reactive wetting and Cassie-Baxter to Wenzel transition. I think this manuscript would be much improved if the authors can identify transitions between Cassie and Wenzel states, and discuss them within the context of reactive wetting, which is present due to the metallic film but not considered by classical wetting models.

Response: In previous studies, the Cassie-Baxter model was applied in systems in which liquid metal generally had gallium oxide on the surface, which hinders metal-metal contact. In our work, we also observed the Cassie-Baxter wetting state before the oxide skin was removed via the HCl vapor treatment. (See Figure R1(a) below) As we pull the EGaIn droplet away, it maintains adhesion only to the top of the posts, indicating that it did not fully penetrate between the posts.

Once the gallium oxide is removed by HCl vapor, EGaIn immediately wets the Cu coating on the post-structured surface by reactive wetting. The wetting behavior of EGaIn in Wenzel regime can be observed in the Supplementary Movie 2 and its snapshots (see Figure R1(b) below), which have been newly added to the revised version. We mainly discussed Wenzel wetting of GaLMs because it is the requirement for the imbibition process.

Figure R1. (a) (Review only) Side-view image of EGaln lifted up after contact to the microstructured metallic surface (without HCl treatment). Gallium oxide adheres only to the top of the posts. (b) (Added as Supplementary Fig. 3) The snapshots of imbibition wetting of EGaln on the microstructured metallic surface after HCl treatment, taken from Supplementary Movie 2. The red dotted line indicates the initial boundary in (i) and the red arrows show the propagation direction of imbibition-induced wetting of EGaln.

"This can be numerically explained based on Wenzel's model and imbibition process, which reveals the dimensions of the post microstructures required for imbibition-induced wetting." Similar to the comment above, I do not believe this statement is met by the manuscript.

Response: The dimensionless parameter newly defined, L , should provide the general quantitative criterion for dimensions of post structures for imbibition. We hope that this can now address your comment.

There are three advantages to imbibition-induced wetting given at the end of the manuscript. The first two are already shown in prior works, although the third seems like a novel claim: "Varying the height of the copper-coated posts potentially enables the formation of the GaLM thin film with controlled thickness. Furthermore, this approach reduces the amount of the GaLM required for thin film formation since the pillars occupy part of the film." The second sentence here requires more quantification. Can the authors provide a strain model? This again brings me back to recent works on solid-liquid biphasic formulations and their impacts on the strain sensitivity of liquid metals. As this is a more structured geometry than those prior works, it would be nice if the authors could relate their electromechanical measurements to the prior proposed models.

Response: The amount of EGaln per unit area ($\mu\text{m}^3/\mu\text{m}^2$) required for formation of a film with thickness of $25 \mu\text{m}$ is $\sim 20 \mu\text{m}^3/\mu\text{m}^2$ at $D=l=25 \mu\text{m}$ and $\sim 9 \mu\text{m}^3/\mu\text{m}^2$ at $D=200 \mu\text{m}$, $l=25 \mu\text{m}$. Considering that the volume of $25 \mu\text{m}$ thick EGaln film without the post is $25 \mu\text{m}^3/\mu\text{m}^2$, the amount of EGaln required is reduced by 20% and 64%, respectively. Thus, the amount of EGaln per unit area required for the film formation can be reduced by introducing the post structure.

Regarding the effect of the post on the resistance of the EGaln film, the presence of posts simply decreases the cross-sectional area of the GaLM film and therefore increases the resistance by Pouillet's law (i.e. Resistance $\propto l(\text{length})/A(\text{cross-sectional area})$). For a stretchable conductor made of incompressible materials, the resistance should follow Pouillet's law, i.e. $R/R_0 = (1+\varepsilon)^2$. We confirmed that our result approximately follows the Pouillet's law, which has been added to Fig. 6a of the revised manuscript, as shown below.

Thanks to your helpful suggestion, we could add more quantitative discussion in the conclusion, regarding the reduced amount of EGaln for thin film formation by introducing the post structures and corresponding resistance based on the Pouillet's law. It now reads as below.

(Page 13)

The resistance increases and recovers proportional to the strain, which agrees well with Pouillet's law ($R/R_0 = (1+\varepsilon)^2$) for incompressible materials, where R is resistance, R_0 is the initial resistance and ε is strain⁴⁴.

Fig. 6 Strain-dependent resistance of EGaIn/Cu/PDMS. a Normalized resistance change of EGaIn/Cu/PDMS depending on strain in the range of 0% to 70%. The maximum strain reached before the PDMS broke is 70% (Supplementary Fig. 7). **The red dots are the theoretical values predicted from Pouillet's law.** **b** The conductivity stability test of EGaIn/Cu/PDMS during repetitive stretching–releasing cycles. The cycle test utilized 30% strain.

(Page 15)

Furthermore, this approach reduces the amount of the GaLM required for thin film formation since the pillars occupy part of the film.

For example, by introducing an array of posts with 200 μm diameter (with 25 μm spacing between them), the volume of the GaLM required for the film formation ($\sim 9 \mu\text{m}^3/\mu\text{m}^2$) reduces by up to 64%, compared to that for the film without posts ($25 \mu\text{m}^3/\mu\text{m}^2$). In this case, however, it needs to be considered that the theoretical resistance estimated by the Pouillet's law also increases 9 times.

Overall, I enjoyed reading this manuscript and I commend the authors for the presentation of their results. I believe this manuscript will make a nice contribution to the literature, though I also believe it is lacking critical analysis and is missing the opportunity to provide new generalized relationships between surface composition, texture, and liquid metal wetting behaviors.

Response: We really appreciate reviewer's comments and helpful advice. We have added several things to improve the analysis. We investigated the effect of surface composition on the imbibition-induced wetting and added the results and the related discussion, which is shown below.

To investigate the influence of the surface composition on wetting behavior, we measured the contact angles and imbibition behavior on the binary surfaces of Si and Cu (Si, Si/Cu (50%/50%), and Si/Cu (25%/75%)). On the surfaces of Si 100% and Si/Cu (50%/50%), the contact angles are $\sim 160^\circ$, similar to those on non-metallic surfaces reported¹⁻³. On the copper rich flat substrate (Cu 75%/Si 25%), the contact angle decreases down to 80° . However, this contact angle (i.e. θ_{eq}) corresponds to 0.43 of L_c , which is still smaller than $L = 1$ for posts with $D = l = H = 25 \mu\text{m}$. Therefore, imbibition is not expected due to pinning. In fact, imbibition does not occur on all of the post-patterned substrates of the binary surfaces tested, which is in a good agreement with the numerical calculation. Thus, the contact angle (i.e. θ_{eq}), and therefore L_c , depend on the chemical compositions of surfaces, determining whether imbibition occurs or not when microstructures are introduced.

1. Kim, D. et al. Hydrochloric acid-impregnated paper for gallium-based liquid metal microfluidics. *Sens. Actuators, B* **207**, 199-205 (2015).
2. Li, G. et al. PDMS based coplanar microfluidic channels for the surface reduction of oxidized Galinstan. *Lab Chip* **14**, 200-209 (2014).
3. Kim, D. et al. Recovery of nonwetting characteristics by surface modification of gallium-based liquid metal droplets using hydrochloric acid vapor. *ACS Appl. Mater. Interfaces* **5**, 179-185 (2013).

(Page 9)

The wetting property can be determined according to the surface composition of substrate. We investigated the effect of surface composition on the wetting and imbibition of EGaIn by co-depositing Si and Cu onto the posts and flat surfaces (Supplementary Fig. 6). As Cu content of the flat, Si/Cu binary surfaces increases from 0 to 75%, the contact angle of

EGaIn decreases from $\sim 160^\circ$ to $\sim 80^\circ$. For the surface of 75% Cu/25% Si, θ_0 is $\sim 80^\circ$, which corresponds to L_c is 0.43 according to the above definition. Since L for the posts with $l = H = 25 \mu\text{m}$ is 1 which is larger than the threshold value L_c , imbibition does not occur on the post-patterned 75% Cu/25% Si surface due to pinning. As the contact angle of EGaIn increases with the addition of Si, higher H or lower l is required to overcome the pinning and for imbibition to occur. Thus, since the contact angle (i.e. θ_0) depends on the chemical compositions of surfaces, it can also determine whether imbibition occurs or not on microstructures.

Supplementary Fig. 6 | Wetting properties of EGaIn on Cu/Si binary surfaces without and with post patterns ($D = l = H = 25 \mu\text{m}$, $L = 1$) in the presence of HCl vapor. Scale bars are 1 mm.

Reviewer #2 (Remarks to the Author):

This work reports the spontaneous selective wetting phenomena of liquid metal on a micro-structured copper substrate in the presence of HCL vapor. And effects of the microstructures to enhance the EGaIn wetting were tested. The article can be of active reference for the society. It is recommended for acceptance after further improvement.

Response: We appreciate the reviewer’s positive evaluation and helpful comments.

Some specific comments:

1. *The originality of the method can be further improved especially compared with those existing similar endeavors.*

Response: Our study presents a simple approach to form flat thin films of GaLMs with controlled geometry using reactive wetting enhanced by imbibition. The wetting process is relatively fast and does not require external force, compared to other previously reported ones. The table below is prepared to compare some important factors between the coating and patterning processes. Moreover, we investigated the effect of the dimensions of the post patterns on the imbibition and presented the quantitative criteria for the spontaneous wetting.

Table R1. Comparison of studies on coating and patterning of GaLMs on metallic substrate (Review only)

Work	Liquid metal	Substrate	Deposition method of liquid metal	External force	Minimum contact angle ¹⁾	Required time for minimum contact angle	Selective wetting
[1]	Ga	Au/PDMS	Thermal evaporation	Required (vacuum and thermal energy)	0°	— ²⁾	×
[2]	Ga	Au/PDMS (micro-structured)	Thermal evaporation	Required (vacuum and thermal energy)	0°	— ²⁾	×
[3]	Galinstan	Cu/SEBS	Direct contact (rolling)	Required (mechanical energy)	0°	— ²⁾	○ by metal-metal interaction ³⁾
[4]	Galinstan	Au/PDMS	Direct contact (rolling)	Required (mechanical energy)	0°	— ²⁾	○ by metal-metal interaction ³⁾
[5]	EGaIn	Porous Cu foam	Electrochemically enabled reactive wetting	Required (electrical energy)	Probably 0°	— ²⁾	×
[6]	EGaIn Galinstan	In foil Sn foil	Spontaneous wetting	Not required	10° 20°	8.2 s 4 days	×
[7]	Galinstan	Cu sheet treated with CuCl ₂	Spontaneous wetting	Not required	4.3°	200 s	○ by metal-metal interaction ³⁾
This work	EGaIn	Cu/PDMS (micro-structured)	Spontaneous wetting enhanced by imbibition	Not required	0°	~ 40 s	○ directed by microstructure

[1] Hirsch, A. et al. Intrinsically stretchable biphasic (solid-liquid) thin metal film. *Advanced materials* **28**, 4507-4512 (2016).

[2] Hirsch, A. et al. A Method to form smooth films of liquid metal supported by elastomeric substrate. *Advanced science* **5**, 1800256 (2018).

[3] Zhu, Z. et al. Fully solution processed liquid metal features as highly conductive and ultrastretchable conductors. *NPJ Flexible Electronics* **5**, 25 (2021).

[4] Li, G. et al. Selectively plated stretchable liquid metal wires for transparent electronics, *Sensors and actuators B: Chemical* **221**, 1114-1119 (2015).

[5] Ma, J. et al. Electrochemically enabled manipulation of gallium-based liquid metals within porous copper. *Materials Horizons* **5**, 675-682 (2018).

[6] Kramer, K. R. et al. Effect of Microtextured surface topography on the wetting behavior of Eutectic gallium-indium alloys. *Langmuir* **30**, 533-539 (2014).

[7] Lin, W. et al. Ultrastrong spontaneous surface wetting of room temperature liquid metal on treated metal surface. *Advanced Materials interfaces* **8**, 2100819 (2021).

¹⁾ 0° means a “complete wetting” was achieved.

²⁾ Not applicable. The required time strongly depends on the deposition process condition. The required times in this table have been discussed mainly for the cases of spontaneous wetting.

³⁾ The selective wetting occurs along the pre-formed metal patterns via metal-metal interaction.

2. The liquid droplets on EGaIn are water. The authors may want to further explain whether there is any impact when used, and how to remove the water.

Response: We have not observed any significant impact of the water during the experiment. The water can be removed by a simple heating process. Reflecting this, we revised our manuscript below.

(Page 11)

In c-e, the small droplets on the surface after imbibition are water through the reaction between gallium oxide and HCl vapor. No significant effect of water formation on the wetting was observed. The water could be easily removed by a simple drying process.

3. In this work, the article used normalized resistance. What's the resistance values of EGaIn/Cu before stretched and after relaxed? Compared them with that of Cu or EGaIn.

Response: The resistivities of EGaIn/Cu before stretched and after relaxed, calculated from the measured resistance, are $8.25 \times 10^{-5} \Omega \cdot \text{cm}$ and $8.9 \times 10^{-5} \Omega \cdot \text{cm}$. These values are higher than, but have the same order of magnitude as, that of pure EGaIn ($\text{EGaIn} \sim 2.9 \times 10^{-5} \Omega \cdot \text{cm}$ and $\text{Cu} \sim 1.7 \times 10^{-6} \Omega \cdot \text{cm}$). The higher resistivity may be due to the voltage drop in the leads and contacts.

4. Besides, whether the 10% variation of resistance value after 4000 stretching cycles is due to the cracks in the rigid copper film? Please give some explanation.

Response: Although the Cu film cracking may contribute to the increase in resistance, the Cu is only ~1% (by volume) of the conductive path. Other possible explanations may be subtle changes to the geometry, changes in contact resistance, or oxidation of the metal (see ref 47). We plan to further investigate if there is any change in morphology and physical/chemical properties of EGaIn, copper coating, and the PDMS substrate during the stretching cycles as a future work.

(Page 13)

As shown in Fig. 6b, the resistance value is maintained after 4000 stretching cycles within 10% variation, which might be due to the continuous formation of the oxide skin during the repetitive stretching cycles⁴⁷.

Reviewer #3 (Remarks to the Author):

The manuscript describes the imbibition-induced, spontaneous and selective wetting of EGaIn on metallized surfaces with microscale topographic features. The manuscript is well written and contains noteworthy results for the flexible electronics community.

Response: We appreciate the reviewer’s positive evaluation and helpful comments.

The following are a few comments to the authors:

Line 119: “this phenomenon is called imbibition”: The authors could add a reference here.

Response: We added the following reference:

Bico, J., Thiele, U. & Quere, D. Wetting of textured surface. Colloids Surf., A 206, 41-46 (2002)

Line 157: The equation could be misread and this reviewer suggests using a second bracket for all of the denominator.

Response: We revised the equation to clearly identify the numerator and denominator, accordingly.

Line 163: “Effect of the width of posts”: Width of posts may be misleading here, because it could mean the diameter of the posts rather than the overall width of rows of posts. The authors may consider rephrasing.

Response: We appreciate this comment. As shown below, we added the mark of “Width (w)” in Fig. 3a and revised the caption to avoid confusion.

(Page 9)

Fig. 3 Effect of the width of rows of posts of Cu/PDMS on the imbibition of EGaIn. **a** EGaIn droplets sitting on the post-patterned Cu/PDMS with different widths of the patterns (w) in air (before exposure to HCl vapor). From the top, the numbers of the rows of posts are 101 (w = 5025 μm), 51 (w = 2525 μm), 21 (w = 1025 μm), and 11 (w = 525 μm). **b** Directional wetting of EGaIn in (a) 10 min after HCl vapor exposure. **c, d** Wetting of EGaIn on the Cu/PDMS with the (c) two rows (w = 75 μm) and (d) one row (w = 25 μm) of the post patterns. The images were captured 10 min after the HCl vapor exposure. The scale bars in (a, b) and (c, d) are 5 mm and 200 μm, respectively. The arrow in (c) indicates the curvature of the “head” of EGaIn through imbibition.

Line 234 and Figure 6: Looking at Figure 6, it appears that the gauge factor is strain dependent over the tested strain range with the gauge factor increasing with increasing strain. If this is true, what is the reason for this? Also, should the gauge factor be defined in the manuscript?

Response: We thank for the review’s comment. We found that the stretchable conductor follows Poillet’s law, which simply means that geometric changes due to stretching change the resistance. We defined the gauge factor and described why the gauge factor at 70% strain increased as follow.

(Page 13)

Gauge factor (GF) denotes the sensitivity of the sensors, which is defined as the ratio of change in electric resistance to the change in strain⁴⁵. The gauge factor increases from 1.7 at 10% strain to 2.6 at 70% strain due to geometric changes to the metal.

Line 249, Discussion: The discussion mentions the testing of pyramid-patterned metallic surfaces, even though the manuscript largely focuses on post-patterned metallic surfaces (besides Fig. 1a). The authors should either limit the discussion to the post-patterned structures or include results on the pyramid-patterned surfaces as well.

Response: The pyramid-pattern was also adopted to confirm that such an imbibition process occurs not only on the post-structures but also on other microstructures with different shapes. For the pyramid-patterns, however, it is not easy to vary the topography dimension of the 3-D shapes and it could be more complex to establish the numerical process for the imbibition. Therefore, we mainly focused on the post-patterned structure. We include the result for the pyramid-patterned surface to confirm that the imbibition is not limited to the post-structured surface.

The manuscript nicely highlights the upper limits for the post geometry. What about the other end of the spectrum; are there any limitations with respect to the smallest post geometries that could be considered?

Response: We appreciated the reviewer's comment. We found that under the condition of $L = l/H < L_c$, imbibition theoretically will continue to occur in very small dimensions. (L and L_c are dimensionless parameters newly defined in the revised version. They provide the criterion of post geometry required for imbibition.) The technical issue for fabrication of the micro-posts would determine the minimum limitation of the post dimension. Inspired by your comments and those of reviewer #1, we investigated imbibition conditions depending on the diameter and interval of posts and further discussed the correlation of imbibition with the dimension of post patterns. The results were added to the revised manuscript and the supplementary data.

(Page 8)

A dimensionless parameter, $L = l/H$, is defined for judging whether or not imbibition occurs. For imbibition, L should be smaller than threshold criterion, $L_c = 1/\{(\sqrt{2} - 1) \tan \theta_{eq}\}$. For EGaIn on copper substrate ($\theta_0 = 25^\circ$), L_c is 5.2. Since L is 8 for 200 μm posts, which is larger than the L_c value, the imbibition of EGaIn does not occur. For further validation of the effect of geometry, we observed imbibition for various H and l (Supplementary Fig. 5 and Supplementary Table 1). The result showed good agreement with our calculation. Thus, L appears to be an effective predictor of imbibition; liquid metal stops undergoing imbibition through the pinning when the distance between posts is relatively large compared to the height of posts.

(Supplementary data)

Supplementary Fig. 5 | Imbibition-induced wetting behavior of EGaIn depending on the dimensions of posts.

Supplementary table 1. θ_c , $\theta_{c,pin}$, L_c and L as a function of the dimensions of posts.

D (μm)	l (μm)	H (μm)	θ_0	θ_c	$\theta_{c,pin}$	L_c	L (l/H)	Imbibition
200	25	25	$\sim 25^\circ$	57°	68°	5.2	1	○
25	25	10	2.5	44°	44°	4.4	2.5	○
25	200	25	8	16°	17°	16	8	×
25	25	1	25	16°	5°	16	25	×

The authors should discuss if there are any requirements with respect to the conformity of the metal film on the 3D PDMS patterns? As an example, how well were the sidewalls of the PDMS coated with metal and how critical is that for the imbibition-induced wetting process?

Response: We agree to review's comment. The conformal deposition of the metal film may be important to the reactive wetting and imbibition. Through the SEM and EDS measurement, we confirmed the conformity of the copper film on both of the top and the sidewall of the PDMS posts. The result was added to the revised manuscript and the supplementary data.

(Page 6)

It was confirmed that the micro-structured surfaces of PDMS substrates were conformally coated with copper (Supplementary Fig. 2).

(Supplementary data)

Supplementary Fig. 2 | Conformality of the copper deposition. **a** Side-view scanning electron microscope (SEM) image of the copper-sputtered PDMS posts. **b** Spatial distribution of copper in **a** by energy dispersive X-ray spectroscopy (EDS) analysis.

REVIEWER COMMENTS

Reviewer #2 (Remarks to the Author):

The authors overall addressed my comments. I am OK with the manuscript and would like to recommend it to be considered by the journal.

Reviewer #3 (Remarks to the Author):

The reviewers have sufficiently addressed the comments and suggestions of this reviewer. Thank you. No further suggestions and comments.